# New CTX-M Group Conferring β-Lactam Resistance: A Compendium of Phylogenetic Insights from Biochemical, Molecular, and Structural Biology

**DOI:** 10.3390/biology11020256

**Published:** 2022-02-07

**Authors:** Jacinta Mendonça, Carla Guedes, Carina Silva, Sara Sá, Marco Oliveira, Gustavo Accioly, Pilar Baylina, Pedro Barata, Cláudia Pereira, Ruben Fernandes

**Affiliations:** 1LABMI—Laboratory of Medical and Industrial Biotechnology, 4200-374 Porto, Portugal; jacinta_m97@hotmail.com (J.M.); carlaguedes015@gmail.com (C.G.); cis@ess.ipp.pt (C.S.); saravanessasa@gmail.com (S.S.); moliveirall97@gmail.com (M.O.); rfernandes@ess.ipp.pt (R.F.); 2ESS—Escola Superior de Saúde, IPP—Porto Polytechnic Institute, 4200-072 Porto, Portugal; gustavomaccioly@gmail.com; 3i3S—Metabesity Research Team, Instituto de Investigação e Inovação em Saúde, 4200-135 Porto, Portugal; 4UVIGO—Facultade de Biología, Universidade de Vigo, 36310 Pontevedra, Spain; 5FMUP—Faculdade de Medicina, Universidade do Porto, 4200-319 Porto, Portugal; 6ESB—Escola Superior de Biotecnologia, Universidade Católica Portuguesa, 4169-005 Porto, Portugal; 7UFP—Faculdade de Ciências da Saúde, Universidade Fernando Pessoa, 4200-253 Porto, Portugal

**Keywords:** CTX-M β-lactamases, extended-spectrum β-lactamases (ESBL), CTX-M-151 new group

## Abstract

**Simple Summary:**

CTX-M β-lactamases are a growing group among extended-spectrum β-lactamases (ESBLs), not only in number, diversity, and functional kinetics. Until now, it has been well-accepted five main clusters within CTX-M β-lactamases. These canonical clusters are CTX-M-1, CTX-M-2, CTX-M-8, CTX-M-9, and CTX-M-25. In the present study, we propose a new sixth cluster, CTX-M-151.

**Abstract:**

The production of extended-spectrum β-lactamases (ESBLs) is the main defense mechanism found in Gram negative bacteria. Among all the ESBLs, the CTX-M enzymes appear as the most efficient in terms of dissemination in different epidemiological contexts. CTX-M enzymes exhibit a striking plasticity, with a large number of allelic variants distributed in several sublineages, which can be associated with functional heterogeneity of clinical relevance. This observational analytical study provides an update of this family, currently with more than 200 variants described, from a phylogenetic, molecular, and structural point of view through homology in amino acid sequences. Our data, combined with described literature, provide phylogenetic and structural evidence of a new group. Thus, herein, we propose six groups among CTX-M enzymes: the already stablished CTX-M-1, CTX-M-2, CTX-M-8, CTX-M-9, and CTX-M-25 clusters, as well as CTX-M-151 as the new cluster.

## 1. Introduction

Penicillin, discovered by Alexander Fleming in 1928, represented a turning point for humanity, revolutionizing Medicine by significantly reducing the number of deaths from bacterial infections worldwide [1,2,3]. However, in 1940, a bacterial penicillinase was identified, foreshadowing the clinical findings that have been occurring since then until today [3,4]. The introduction of antimicrobial agents in clinical practice was followed by the gradual appearance of resistant bacterial strains, whose threat of dissemination constitutes a worrying reality in Public Health [4,5]. In this matter, it is estimated that approximately 25,000 patients die each year in the European Union because of infections caused by multidrug-resistant bacteria, and the associated costs are estimated circa 1.5 billion euros/per year) [6].

The β-lactams are widely used antibacterial antibiotics in therapy, due to their high efficacy, low toxicity, and pharmacokinetic properties [1,4,7]. Subsequently, the production of β-lactamases is the predominant cause of resistance to β-lactam antibiotics in Gram-negative bacteria [1,8]. These enzymes cleave the amide bond in the β-lactam ring, rendering β-lactam antibiotics harmless to bacteria [9,10,11]. The introduction of expanded spectrum cephalosporins in clinical practice in the early 1980s represented a breakthrough for the treatment of infections caused by Enterobacteriaceae and other Gram-negative pathogens [12].

As a result, the massive use of expanded-spectrum cephalosporins generated a selective pressure that was followed by the rapid emergence of new β-lactamases, degrading and conferring resistance to these compounds, named extended-spectrum β-lactamases (ESBLs). Among all ESBLs, the CTX-M type β-lactamases have proved to be, by far, the most successful in disseminating in the clinical setting and have overall become the most prevalent ESBLs worldwide [13,14,15,16,17].

The CTX-M-type β-lactamases belong to a quite heterogeneous lineage of molecular class A active site-serine β-lactamases (Ambler classification), and according to the functional classification scheme of Bush et al., they are clustered in group 2be [18,19].

At present, the CTX-M family comprises about 200 enzymes, clustered in at least five sublineages or groups (CTX-M-1, CTX-M-2, CTX-M-8, CTX-M-9, and CTX-M-25). Each group, in turn, includes a few minor allelic variants that differ from each other by one or few amino acid substitutions. This molecular heterogeneity assumes clinical relevance, since it may be responsible for its resistance profile [20,21]. Moreover, there are at least three variants from CTX-M-1 (namely, CTX-M-64, CTX-M-123, and CTX-M-132) and one variant from CTX-M-9 (Toho-9) that exhibit a hybrid structure [20]. Recently, CTX-M-151 was described as possessing a low degree of amino acid conservation when compared with the other five CTX-M groups (63.2% to 69.7% identity) and authors speculate that this could even be a new group within the CTX-M broad family [22].

The CTX-M ESBL were initially reported in the second half of the 1980s, coming from chromosomal genes resident in members of the genus *Kluyvera,* which includes several environmental species with little or no pathogenic activity against humans [23,24,25,26,27]. Precursors of genes encoding enzymes of the CTX-M-1 group have been detected in strains of *K. cryocrescens* and *K. ascorbate;* the CTX-M-2 group in *K. ascorbate;* and, finally, the CTX-M-8, CTX-M-9, and CTX-M-25 groups occur in *K. georgiana* [20,28,29,30,31,32,33,34]. In this matter, there are a few review articles that have been published in the past [20,21,23,24,28,35,36].

Considering that the epidemiology of CTX-M-type ESBLs is evolving rapidly, the objective of this paper is to provide a concise update on CTX-M-type β-lactamases, with a focus on aspects related to enzymatic properties, structural relationships, and phylogenetic significance of the CTX-M-type ESBLs, proposing a new group within CTX-M family.

## 2. Materials and Methods

Through the database of the Bioproject “Bacterial Antimicrobial Resistance Reference Gene Database”, curated by NCBI the GenBank^®^, the accession number of each enzyme was obtained. This number can be consulted in such databases and, from there, all the nucleotide and the protein sequences as well [37,38].

It is important to indicate that the bioproject mentioned above contains annotated sequence records for representative DNA sequences that encode proteins conferring or contributing to resistance to various antibiotics. Thus, the β-lactamase collection was built by aggregating assemblies from multiple sources, curating conflicts, and including additional sequences found by review of the literature. Major contributors to the National Center for Biotechnology Information (NCBI) reference set of antimicrobial resistance (AMR) genes include the Lahey group, Resfinder, the Center for Veterinary Medicine at the Food and Drug Administration, the Comprehensive Antimicrobial Resistance Database (CARD), the Institut Pasteur, Dr. Marilyn Roberts, and Dr. Derrick Crook [37].

Currently, there are 214 CTX-M enzymes belonging to this database. Subsequently, the protein and nucleotide sequences were analyzed through Bioinformatics programs, more specifically: (i) Clustal Omega for sequences alignment; (ii) Jalview for visualization, analysis and comparison of previously aligned sequences; and (iii) NGPhylogeny.fr for the construction of the phylogenetic tree (dendrogram), following the UPGMA (Unweighted Pair Group Method with Arithmetic mean) method, which assumes evolutionary rates for all taxonomic units, meaning it groups the shortest distances and recalculates new distances using arithmetic means for a new grouping [39,40,41]. This bioinformatics analysis allowed all CTX-M variants to be clustered, by comparing their amino acid sequences’ homology and by their phylogenetic relationship. For this purpose, we defined the cut-off value for the phylogenetic identity as ≥95% for the several taxonomic units belonging to the same cluster. Furthermore, the comparative study of protein sequences allowed the measurement of possible mutational hot spots. To this end, it is important to mention that the sequences were compared individually by analogy with the lowest numbered variant within the cluster.

## 3. Results

As previously described, the analysis of the CTX-M enzymes’ phylogeny showed conserved regions of amino acids, further supporting the idea of a common ancestral protein identity. In addition, these amino acids are also the structural requirements of CTX-M β-lactamases, which are necessary for the presence of their typical profiles (phenotypic as well as biochemical).

### 3.1. Phylogenetic Analysis

The phylogenetic trees obtained are presented (Figure 1). Clusters may be observed by colors (in black, the CTX-M-1 cluster; in red, the CTX-M-2; in grey, the CTX-M-8; in orange, the CTX-M-9; in green, the CTX-M-25. In addition, this analysis returned a new and apparently more distant group, the CTX-M-151 enzyme (blue line). Considering these results, a multiple sequence alignment approach was performed to assess the amino acid sequence.

### 3.2. Molecular and Structural Study

The amino acid sequence was explored by means of multiple sequence analysis using Clustal Omega (The EMBL-EBI search and sequence analysis tools APIs, 2019). With this approach, alignments between classic Bonnet representative CTX-M enzyme members included CTX-M-1, CTX-M-2, CTX-M-8, CTX-M-9, and CTX-M-25. Moreover, CTX-M-151 amino acid sequence was included for comparison purposes (Figure 2).

As observed in Figure 2, sequence variability among CTX-M enzymes is extraordinary. There are several nonconserved amino acid substitutions and it is reasonable to expect great impact in the three-dimensional structure of these enzyme, since in most positions amino acid substitutions change its class, e.g., changing from polar amino acids for negatively charged amino acids, the substitution for an aromatic polar amino acid for a positively charged one, and so on. Another important aspect observed in Figure 2 is that CTX-M-151 presents 304 amino acids, unlike the remaining CTX-M enzymes aligned, which present 291 amino acids. In Figure 3, we compare the tridimensional (3D) structure of CTX-M-1, here used as an example of the CTX-Ms with 291 amino acids, with CTX-M-151′s 3D structure, highlighting in red the extra 13 residues. Interestingly, when we look at these enzymes’ structures, their sequences’ variability seems unnoticeable, as they display notorious similarities when overlapped (Figure 3A).

As we can see from Figure 3, when we overlap the respective 3D structures of CTX-M-1 and CTX-M-151, the similarities stand out, despite the extension of amino acid substitutions between both enzymes that account for more than 50% (Figure 3A). The residues corresponding to the Omega loop show two substitutions: Thr168 and Thr171 in CTX-M-1 that correspond to Asn176 and Ser179 in CTX-M-151, respectively (Figure 3B). However, these substitutions do not seem to induce any structural alteration, conserving the Omega loop that is transversal to all CTX enzymes.

### 3.3. CTX-M Clusters

#### 3.3.1. Sequence Mutations in Cluster CTX-M-1

CTX-M-1-like enzymes represent the largest and the most diverse cluster among CTM-M ESBL enzymes. Figure 4 shows the CTX-M-1 gene with all the amino acid substitutions among this cluster mapped, point by point. For example, in the seventh amino acid of this group, two substitutions occur: Arg7His is a substitution present in CTX-M-194 and Arg7Ser is present in both CTX-M-162 and CTX-M-163.

Five major hot spots are found in the cluster (M1 to M5). These hot spots include five key substitutions: Val80Ala, Asp117Asn, Ser143Ala, Asp242Gly, and Asn289Asp. In Table 1 is listed the CTX-M-1-like enzymes by their enzyme index. For example, Val80Ala may be seen in indexes 3, 10, 12, 15 … 232. This means that mutation Val80Ala is also present in CTX-M-3, CTX-M-10, CTX-M-12, CTX-M-15, and CTX-M-232 among all others listed in Table 1.

All 106 enzymes belonging to this group present a protein sequence with 291 amino acids, except CTX-M-211 (with 294 amino acids) where the insertion of three amino acids at the beginning of the sequence (methionine, valine, lysine) is observed.

Through the observation of Figure 4, it is possible to verify that the enzymes in this group suffer more than one mutation, with the exception of CTX-M-23 (M3-Ser143Ala), CTX-M-36 (M1-Val80Ala), CTX-M-58 (Pro170Thr), CTX-M-61 (M5-Asn289Asp), CTX-M-146 (Lys237Arg), CTX-M-158 (Ala127Val), CTX-M-166 (Ala123Val), CTX-M-175 (Ala123Thr), and CTX-M-222 (Val151Ile). Furthermore, only the mutations that occur in CTX-M-158, CTX-M-166, CTX-M-175, and CTX-M-222 enzymes are variant-specific.

Subsequently, although having more than one amino acid mutation, some enzymes differ by the occurrence of one or more variant-specific mutations. These are listed below in Table 2.

Additionally, as it can be observed, five mutations occur in many variants (hotspots). As such, they have been identified from M1 to M5. Val80ala (M1) substitution occurs in 76 distinct variants, Asp117asn (M2) occurs in 91, Ser143ala (M3) occurs in 93, Asp242gly (M4) occurs in 66, and Asn289asp (M5) occurs in 88 enzymes.

Among the variants that do not have exclusive mutations, the following ones have a differentiated combination between M1 to M5 substitutions: CTX-M-3 (M1, M2, M3, and M5); CTX-M-15 (M1, M2, M3, M4, and M5); CTX-M-22 (M1, M2, and M3); CTX-M-28 (M1, M2, M3, and M4); CTX-M-57 (M2, M3, M4, and M5); CTX-M-79 (M2, M3, and M4); CTX-M-116 (M2 and M3) and CTX-M-136 (M2, M3, and M5).

All other enzymes have—besides the combination of the substitutions M1 to M5—others that also occur in more than one variant.

It should also be noted that the most distant protein variant of CTX-M-1 is CTX-M-199, accounting for 30 different amino acid substitutions, followed by CTX-M-64 with 29 substitutions (28 of which are shared with CTX-M-199), CTX-M-123 with 20 substitutions, and CTX-M-132 with 15 replacements. This is extremely relevant from a confirmatory point of view, corroborating the hypothesis of the existence of hybrid enzymes considered as subgroups.

In the group of amino acid substitutions known for having an impact in the resistance profile of enzymes, the following are verified: Ser133Gly (in the CTX-M-189 enzyme); Pro170Ser (in the CTX-M-52 and CTX-M-62 enzymes); and Asp242Gly.

#### 3.3.2. Sequence Mutations in Cluster CTX-M-2

CTX-M-2-like enzyme are much less diverse than the CTX-M-1 cluster. Nevertheless, CTX-M-2 constitutes the third most diverse group within CTX-M ESBL enzymes. Figure 5 illustrates all the amino acid mutations that occur within CTX-M-2 cluster’s members, when compared with the CTX-M-2 sequence.

As it can be verified, the mutations are quite punctual, occurring in low number as well as in reduced sequence positions. Thus, the vast majority of the amino acid substitutions are exclusive to a certain variant, contrary to what has been observed in the CTX-M-1 group. In addition, more than half of the variants in this group result from a single amino acid substitution, namely, CTX-M-20 (Ile279Phe), CTX-M-31 (Thr31Ser), CTX-M-35 (Pro170Ser), CTX-M-44 (Ser275Arg), CTX-M-56 (Ser275Asn), CTX-M-59 (His92Leu), CTX-M-74 (Pro170Thr), CTX-M-75 (Pro18Ser), CTX-M-92 (Ala208Glu), CTX-M-97 (Arg7Gly), CTX-M-131 (Asp242Gly), CTX-M-141 (Ala208Glu), CTX-M-165 (Asn31Ser), CTX-M-171 (Arg97Ser), and CTX-M-200 (Ser240Ile).

The following are variants that have substitutions other than variant-specific ones: CTX-M-4 (Leu58Gln and Leu288Met), CTX-M-5 (Glu255Ala), CTX-M-6 (Arg64Leu), CTX-M-7 (Glu124Gln), CTX-M-76 (Val233Gly), CTX-M-77 (His289Arg), CTX-M-95 (Val83Leu), and CTX-M-115 (Gly290Ser). Additionally, the CTX-M-229 enzyme has not only two individualizing amino acid substitutions (Ile176Ala and Arg165Ser), but also the deletion of an alanine at position 175.

This analysis also allows us to observe that two variants do not have specific mutations (the CTX-M-43 and the CTX-M-124 enzymes). The CTX-M-43 enzyme has replacements occurring simultaneously in CTX-M-44 and CTX-M-131. The CTX-M-124 replacements occur in positions 251 and 279.

It is possible to ascertain that the variants that have undergone more mutations compared with the CTX-M-2 sequence are the CTX-M-4 and the CTX-M-7 enzymes, with eight substitutions. Of these, six occur simultaneously in both variants (in positions 64, 101, 102, 128, 174, and 223 of the sequence).

Furthermore, we verified the existence of a substitution (Ile279Val) that occurs in eight distinct enzymes (CTX-M-5, CTX-M-6, CTX-M-7, CTX-M-76, CTX-M-77, CTX-M-95, CTX-M-115, and CTX-M-124).

Regarding the substitutions that interfere in the avidity by the substrate, the following are verified in this group: Pro170ser (in CTX-M-35) and Asp242gly (in CTX-M-43 and CTX-M-131 enzymes).

#### 3.3.3. Sequence Mutations in Cluster CTX-M-8

The CTX-M-8-like cluster is a very small group composed, so far, by only 14 members. Thus, only a few substitution sites have been found in this cluster and include amino acid positions 1, 92, 112, 181, and 275. The diagram in Figure 6 represents all the amino acid mutations that occur between CTX-M-8 cluster members, compared with the CTX-M-8 sequence.

This group is composed by the CTX-M-8, CTX-M-40, and CTX-M-63 enzymes. Therefore, the analysis of protein sequences was performed by comparing their sequences to the CTX-M-8 sequence. As such, we obtained the diagram present in Figure 6. We observe that the number of amino acid substitutions is quite small (only four substitutions). It is also noted that, except for Ser274Asn that occurs in the CTX-M-63 enzyme, all other substitutions occur simultaneously in both variants.

Additionally, a methionine deletion is observed at position 1 in the CTX-M-40 sequence, as well as in the CTX-M-63 sequence. For this reason, in this group, only the CTX-M-8 enzyme has 291 amino acids, considering that the other variants have minus one amino acid.

#### 3.3.4. Sequence Mutations in Cluster CTX-M-9

Figure 7 characterizes all the amino acid mutations that occur between CTX-M-9-like members compared with the CTX-M-9 sequence.

When analyzing Figure 7, it appears that most of the enzymes belonging to this group suffer more than one amino acid mutation. However, there are five variants that only have an amino acid substitution, they are CTX-M-14 (Ala234Val), CTX-M-16 (Asp242Gly), CTX-M-44 (Gly46Arg), CTX-M-51 (Ala80Val), and CTX-M-214 (Ala112Thr). It is important to mention, however, that these mutations are not exclusive to these enzymes and occur in many others, as verifiable in Figure 7.

Additionally, it is possible to verify that there is a substitution (Ala234Val) that occurs in a greater number of different enzymes (it occurs in 59 of the 63 enzymes in this group), not being found only in the CTX-M-16, CTX-M-44, CTX-M-51, and CTX-M-214 variants.

Therefore, all other enzymes in this group have the Ala234Val substitution. Of these, it is possible to verify that some have one or more mutations exclusive to themselves, as presented in Table 3:

It should also be noted that CTX-M 110, in addition to the two substitutions mentioned above, is the only enzyme that presents the insertion of an amino acid at the end of the sequence (insertion of a leucine in position 292).

The remaining enzymes also have more substitutions common to other enzymes, and some even have variant-exclusive mutations. Thus, it appears that, of these, there are 15 enzymes with the Asp242Gly substitution, seven enzymes with Ala80Val, five with Pro170Ser, and seven with Ser275Arg.

Thus, among those with the Asp242Gly substitution, it appears that CTX-M-27 is the only one with only that substitution. Next, CTX-M-93, CTX-M-129, CTX-M-159, CTX-M-174, CTX-M-192, and CTX-M-233 present, in parallel, exclusive variant mutations, respectively, Leu172Gln, Lys215Glu, Glu124Lys, Gln7Leu, Lys237Arg, and Glu160Lys.

Regarding the remaining variants, it is possible to verify that CTX-M-98 also comes from the Ala80Val replacement. The CTX-M-195 and CTX-223 still have a common substitution (Asn109Ser), differing mainly due to the presence of the exclusive substitution of CTX-M-233 (Glu160Lys). CTX-M-121 features, in addition, the Ala112Thr replacement, and CTX-M-196 features the Ser275Arg replacement.

This set of enzymes also includes CTX-M-102 and CTX-M-137. The first has Ala207Glu substitution (common to CTX-M-105) and a leucine deletion at position 291 (concomitant with CTX-M-19 and CTX-M-105) and the second has a set of distinct substitutions, described below.

Still analyzing Figure 7, it is possible to verify that the enzymes that present more mutations in relation to CTX-M-9 are CTX-M-137 and CTX-M-221, accumulating 19 and 21 different substitutions, respectively. Between them, they share 18 amino acid substitutions (Gln195Asn, His200Lys, Glu204Asp, Thr205Ser, Leu214Met, Arg225Gln, Thr230Ala, Thr233Val, Ala234Val, Gln255Lys, Gly256Asp, Val261Ile, Gln271Pro, Asn272Lys, Arg285Lys, Ile287Val, Ala288Thr, and Glu289Asp). Thus, they mainly differ because the CTX-M-137 also has the Asp242Gly replacement, as mentioned above, and another replacement, Val6Met (simultaneous with the CTX-M-13). In turn, CTX-M-221 has a replacement of a proline for a serine at position 170 and a replacement of a threonine with an isoleucine at position 264.

The CTX-M-13 has, as previously described, the replacement Val6Met (common with the CTX-M-137), the Ala157Glu substitution (common with the CTX-M-122), and two other exclusive substitutions (Ala56Lys and Ala57Glu). In parallel, in the set of enzymes with the Pro170Ser substitution—CTX-M-19, CTX-M-99, CTX-M-147, CTX-M-219, and CTX-M-221—CTX-M-99 is the only one that also has the Ser275Arg substitution. This substitution also occurs in the CTX-M-24, CTX-M-65, CTX-M-130, CTX-M-148, and CTX-M-196 enzymes. Of these, it is important to mention that CTX-M-65 also has the Ala80Val substitution, CTX-M-130 the Arg277His substitution, CTX-148 the Ala80Val and Met78Ile substitutions, and CTX-M-196 the Asp242Gly.

In position 275, a serine is replaced by a histidine in CTX-M-125 and CTX-M-215 variants. Thus, the CTX-M-215 also has an Asn135Asp substitution (common to CTX-M-191).

The CTX-M-87 and the CTX-M-90 variants differ from one another in the substitution of a proline for a leucine in position 170 in the CTX-M-87 enzyme.

Analyzing CTX-M-49 and CTX-M-50, it appears that both have the Ala51Pro substitution, but the first additionally features the Gly46Arg (in common with the CTX-M-44). Finally, regarding the substitutions already described as influencers of the resistance pro-file, the following are verified in this group: Ala80Val, Pro170Ser, and Asp242Gly.

#### 3.3.5. Sequence Mutations in Cluster CTX-M-25

The diagram in Figure 8 represents all the amino acid mutations that occur within CTX-M-25-like members when compared with the CTX-M-25 sequence.

The analysis of Figure 8 shows that CTX-M-100 results from the substitution of a valine by an alanine at position 80, CTX-M-217 of an asparagine by a serine at position 109, CTX-M-160 of an isoleucine by a valine at position 106, and CTX-M-89 of a glycine by an aspartate at position 242.

However, Val80ala and Gly242Ser substitutions are not exclusive to the abovementioned variants. They occur together in a substantial number of other variants, they appear in seven and eight different enzymes, respectively.

Therefore, the remaining variants occur by the presence of more than one amino acid mutation, which can be seen in more than one enzyme of this group. For example, CTX-M-91 has two amino acid substitutions, one of which has already been evidenced previously (Gly242Ser) and another variant-exclusive (Ala192Ser). Similarly, it appears that the variant-exclusive substitutions are the following: Ser126Ile (in CTX-M-141); Gln255Arg (in CTX-M-26); Met212Thr (in CTX-M-78); Gly29His and Thr178Ala (in CTX-M-152); Thr157Ala and Arg277Gly (in CTX-M-205). Moreover, CTX-M-152 has a deletion of a methionine at position 1.

It is still possible to infer which enzymes are more distinct from CTX-M-25 from a protein point of view. Thus, CTX-M-185 has 12 amino acid substitutions, followed by CTX-M-78 (with 11 substitutions), CTX-M-205 (with 10 substitutions), and CTX-M-152 (with eight substitutions and one deletion).

## 4. Discussion

### 4.1. Phylogenetic Study

Regarding the phylogenetic studies, Figure 1 and Figure 2 show the CTX-M enzymes clustered in groups, with each group being named based on the lower number index within. Analysis of the phylogenetic results led us to question the existence of six CTX-M groups, and not five as stated until now, as highlighted in Figure 1 by color code with CTX-M-1 highlighted in black color; CTX-M-2 in red; CTX-M-8 in grey; CTX-M-9 in orange; CTX-M-25 in green; and, finally, the first member of the new proposed group, CTX-M-151, represented in blue. To the best of our knowledge, CTX-M-151 is the first and only member of this new group.

Further analysis of Figure 1 showed that, in the CTX-M-9 group, there is a clear separation into two taxonomic units (CTX-M-221 and CTX-M-137), similar to what occurs in the CTX-M-1 group for enzymes CTX-M-199, CTX-M-64, CTX-M-123, and CTX-M-132. These data support the hypothesis of the existence of subgroups within the groups. This unequivocal distancing influences the intragroup identity, thus affecting the similarity percentage of the amino acids shown in Figure 2. As a result, the identity in the CTX-M-1 and CTX-M-9 groups increases approximately from 90% and 92%, respectively, to 97%. In fact, and in agreement with Bonnet [28], it is verified that the CTX-M-9 and CTX-M-1 groups are the most distant ones, in evolutionary terms, being the CTX-M-9 group closer to the common ancestor, and consequently, the CTX-M-1 group further away.

In parallel, it is found that there are 106 enzymes belonging to the CTX-M-1 group, 27 enzymes to the CTX-M-2, three enzymes to the CTX-M-8 group, 63 enzymes to the CTX-M-9, and 14 to the CTX-M-25. Ultimately, these results follow, once again, with complete agreement with the previous phylogenetic grouping proposed by Bonnet [28]. Moreover, it is also noted that, in these results, the CTX-M-151 group consists solely of that ESBL itself. The presence of 13 more amino acids in this enzyme sequence further highlights the singularity of CTX-M-151 among the remaining groups and re-enforces the speculations of a new group among all CTX-M groups described.

### 4.2. Molecular and Structural Study

As previously described, the phylogeny of CTX-M enzymes showed that they share a certain protein identity, supporting the fact that there are critical amino acids that are conserved between the different groups. In addition, these amino acids represent structural requirements of β-lactamases CTX-M, necessary for the presence of their typical profiles (phenotypic as well as and biochemical).

The intercluster molecular and structural study was performed by multiple alignment. The sequence alignment, in Figure 3, demonstrated 291 amino acids in clusters CTX-M-1, CTX-M-2, CTX-M-8, CTX-M-9, and CTX-M-25, while the CTX-M-151 enzyme presented 304 amino acids, being therefore the lengthy protein among CTX-Ms. Amino acid mutations were listed by group. A single diagram was prepared for each CTX-M according to the Bonnet classification [28] with all amino acid mutations known to date.

Several amino acid substitutions occur in the sequences of the CTX-M-2, CTX-M-8, CTX-M-9, and CTX-M-25 enzymes. Some amino acid substitutions appear to be group-specific. Contrariwise, there are some substitutions that may occur in all CTX-M groups. Some examples include Gln8Arg, Thr10Met, Glu39Ala, Arg42Lys, Gln43Ser, Ser86Gln, Pro91Lys, Asp117Asn, Ser143Ala, and Met214Leu. Some of these substitutions result in a change of the amino acid chemical group, and correspond to the substitutions in positions 39, 91, 117, and 143.

CTX-M-2 cluster presents 52 different amino acid modifications compared with CTX-M-1, distributed by 31 hotspots. Nevertheless, most of these substitutions affect only 13 amino acid positions, namely, Lys4Gln, Phe9Ser, Met12Thr, Thr14Met, Pro25Thr, Asp32Ser, Lys36Gln, Ala38Glu, Glu90Asp, Gly203Ala, Ala221Ser, Ala230Lys, and Asp256Asn.

In CTX-M-8 there are about 50 substitutions. However, only eight are exclusive to this enzyme, namely, Lys4His, Ser59Ala, Asn92Lys, Ile93Val, Lys102Ser, Ala112Thr, Glu161Asp, and Lys200Ser. As a follow-up, of all the above, only Ile93val does not correspond to a substitution with alteration of the amino acid grouping.

In the CTX-M-9 cluster, the evidence presents 55 amino acid substitutions, 23 of them occurring exclusively in the enzyme CTX-M-9. Of these, 17 substitutions occur with modification of the amino acid chemical grouping, namely, Arg7Gln, Met12Ala, Thr14Ala, Ala31Ser, Asp32Ala, Ile61Val, Ala65Gly, Arg97Pro, Ser103Ala, Ala149Gly, Ser150Gly, Lys200His, Val233Thr, Lys255Gln, Lys272Asn, Thr288Ala, and Asn289Glu.

Then, in the CTX-M-25 group, there are up to 50 distinct amino acid substitutions, 12 of which are specific to CTX-M-25. Of these, only seven correspond to the chemical alteration of the amino acid (Gly22Ala, Ala123Gln, Thr192Ala, Lys200Asp, Thr212Met, Asp242Gly2, and Pro262Ser).

In addition, the results obtained by comparing the protein sequences confirms that the most divergent group is CTX-M-2. However, the one with the highest number of amino acid substitutions that involves the amino acid chemical combination is the CTX-M-9 group. Therefore, these data corroborate what was previously mentioned regarding the evolutionary proximity of the groups.

For CTX-M-151, the phylogenetic analysis found no other member to this new group; therefore, no diagram was created. CTX-M-151 presents 163 substitutions, when compared with CTX-M-1, representing more than 50% of amino acid substitutions, despite the 3D structural similarities (Figure 3).

### 4.3. CTX-M-151 as a New Lactamases Group

Considering previous speculations regarding this enzyme as a new CTX-M’s group, we took upon the task of phylogeny assessment, as well as molecular structure analysis. Results provided a distinct profile, with 13 more amino acids than the other CTX-M groups and with a total of 163 substitutions when compared to CTX-M-1. Looking into the literature, a recent study conducted by Ghiglione et al. provided further insights regarding this β-lactamase, particularly its pharmacological profile. According to the authors, CTX-M-151 presents higher efficiency hydrolyzing ceftriaxone than ceftazidime (3rd generation of cephalosporins) with 6000-fold higher k_cat_/K_m_ (specificity of an enzyme to a substrate) values when compared with the remaining CTX-Ms groups. Moreover, inhibition by Avibactam showed a deacylation rate (K_off_) up to 14-fold higher than the other class A β-lactamases. Although further experimental studies should be conducted to validate the bioinformatics analysis, these results corroborate our proposal to classify CTX-M-151 as a distinct sixth group among the β-lactamase enzymes [22].

## 5. Conclusions

Since their first appearance, CTX-M enzymes were considered a fast-growing group among class A hydrolytic enzymes against beta-lactams. This class of enzymes, until now divided into five groups, present common aspects regarding antimicrobial profile, such as their ability to easily hydrolyze cefotaxime but poorly ceftazidime [42,43]. Still, in terms of epidemiological dissemination, their mechanism is not yet fully understood. Perhaps it may be related to the fact that CTX-M enzymes exhibit a striking plasticity, with many allelic variants belonging to several sublineages.

To date, five well-defined groups among the CTX-M family have been accepted since the work in 2004 of Bonnet [28]. In the present work, we proposed six groups, with the inclusion of CTX-M-151 as a new group, a continually fast growing group of resistance enzymes. Besides our results, experimental data found in literature further corroborate the need to update the current classification of five CTX-M groups, since CTX-M-151 shows a distinct drug hydrolysis profile when compared with the other CTX-Ms.

On a more global view, the need for continuous research regarding CTX-Ms becomes evident. As the incidence of infections and mortality numbers in clinical practice increases, the resistance profile of these enzymes proves to mutate efficiently over time. Hence, with the rising of new variants with distinct therapeutic responses, the need to deepen the knowledge of β-lactamases biological patterns are urgent [44]. This paper aims to alert for the evidences of a new CTX-M, highlighting the need for further studies regarding this topic.

## Figures and Tables

**Figure 1 biology-11-00256-f001:**
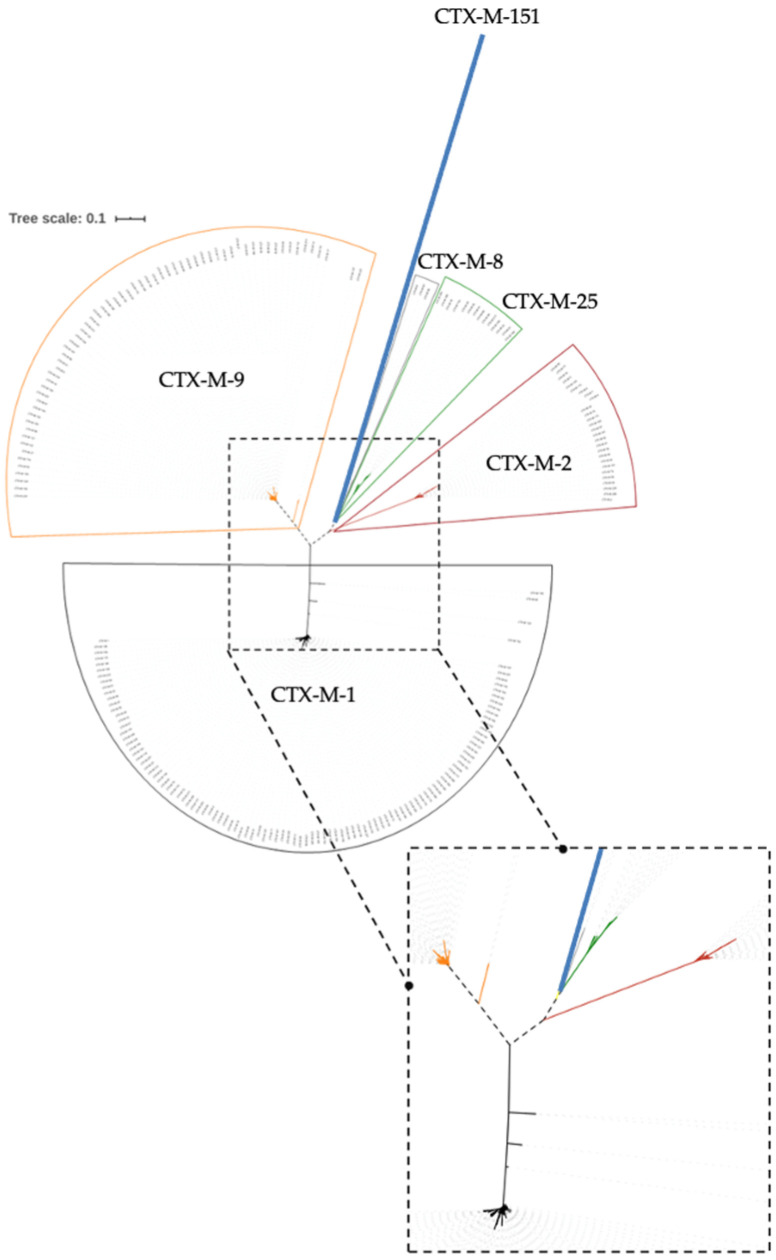
Unrooted phylogenetic tree, in which six different clusters may be observed by colors (in black, the CTX-M-1 cluster; in red, CTX-M-2; in grey, CTX-M-8; in orange, CTX-M-9; in green, CTX-M-25; and in blue, CTX-M-151).

**Figure 2 biology-11-00256-f002:**
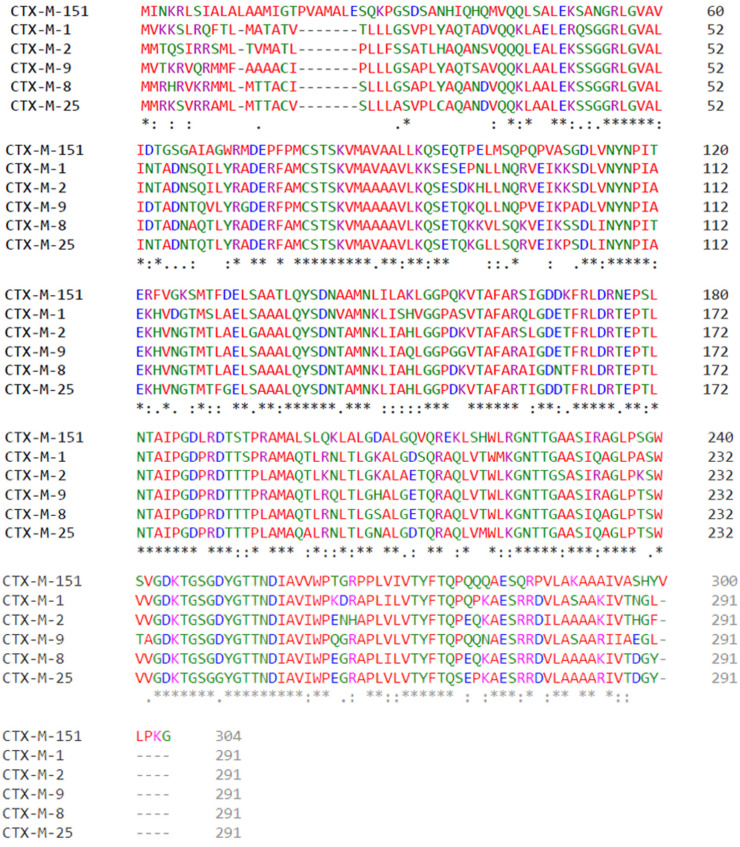
Amino acid multiple alignments. Hydrophobic amino acids are represented in red (A, I, L, M, F, W, V, P). Polar amino acids are represented in green (N, Q, S, T, C, G, H, Y). Positively charged amino acids are represented in magenta (K, R). Negatively charged amino acids are represented in blue (D, E). CTX-M-1, CTX-M-2, CTX-M-8, CTX-M-9, and CTX-M-25 have 291 amino acids, while CTX-M-151 has 304 amino acids. “*” represents fully conserved residues, “.” indicates conservation between groups of strongly similar properties and “:” indicates conservation between groups of weakly similar properties.

**Figure 3 biology-11-00256-f003:**
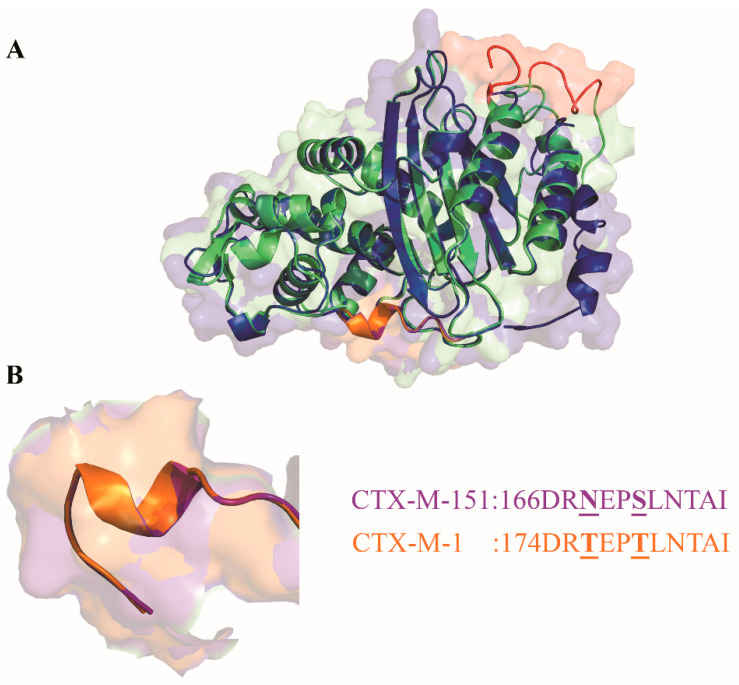
(**A**): Overlap of CTX-M-1 (dark blue) and CTX-M-151 (light green) three-dimensional (3D) structures. In red, residues highlight the extra 13 amino acids of CTX-M-151 in comparison with CTX-M-1. (**B**): The Omega loop (active site), common to all CTX-M enzymes, is represented in orange for CTX-M-1 and in purple for CTX-M-151, with their respective sequences.

**Figure 4 biology-11-00256-f004:**
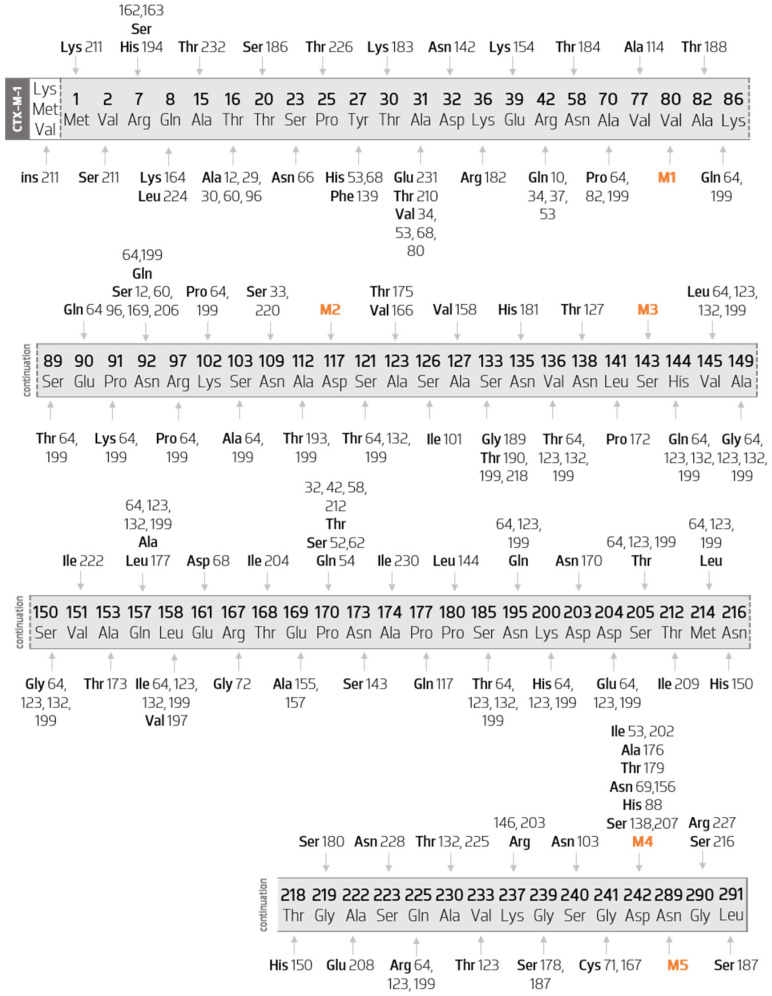
Diagram of the several mutations of the members of CTX-M-1 cluster when compared with the CTX-M-1 sequence. Scheme based on available sequences and phenotype in Bacterial Antimicrobial Resistance Reference Gene Database, in which five major hotspots are represented by Val80Ala (M1), Asp117Asn (M2), Ser143Ala (M3), Asp242Gly (M4), and Asn289Asp (M5).

**Figure 5 biology-11-00256-f005:**
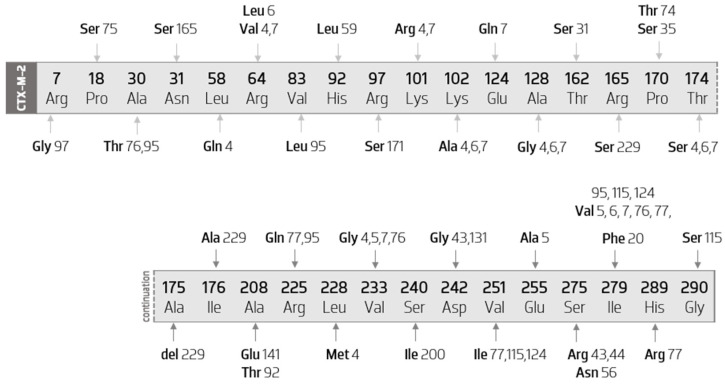
Diagram of the mutations of CTX-M-2 cluster’s members, compared with the CTX-M-2 sequence. Scheme based on available sequences and phenotypes in the Bacterial Antimicrobial Resistance Reference Gene Database.

**Figure 6 biology-11-00256-f006:**
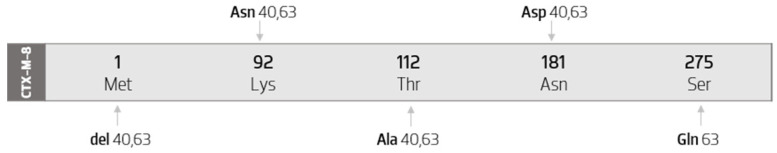
Diagram of the mutations of CTX-M-8 cluster’s members, compared with the CTX-M-8 sequence. Scheme based on available sequences and phenotype in the Bacterial Antimicrobial Resistance Reference Gene Database.

**Figure 7 biology-11-00256-f007:**
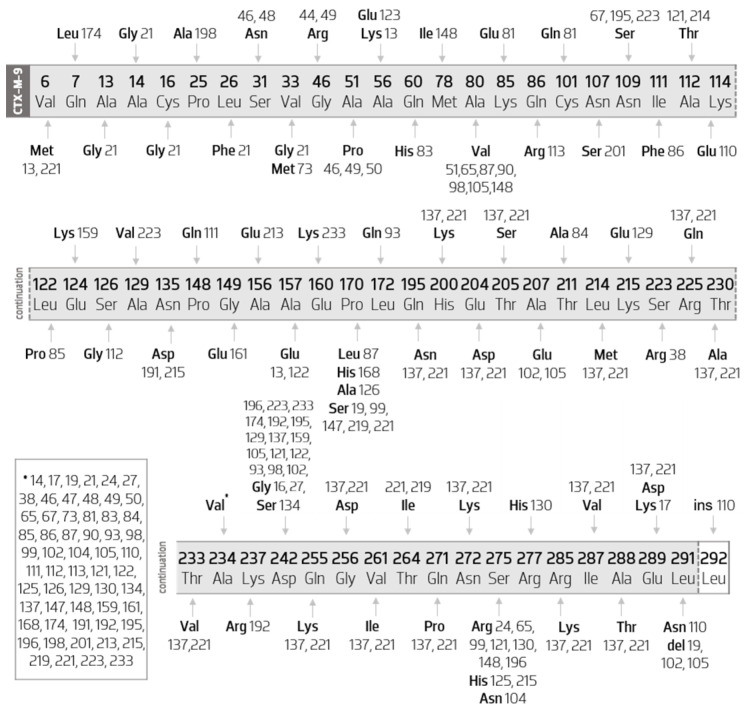
Diagram of the mutations of CTX-M-9-like members compared with the CTX-M-9 sequence. Scheme based on available sequences and phenotype in the Bacterial Antimicrobial Resistance Reference Gene Database. * indicates all the indexes of the CTX-M-9 like members that present a substitution of an alanine by a valine at position 234.

**Figure 8 biology-11-00256-f008:**
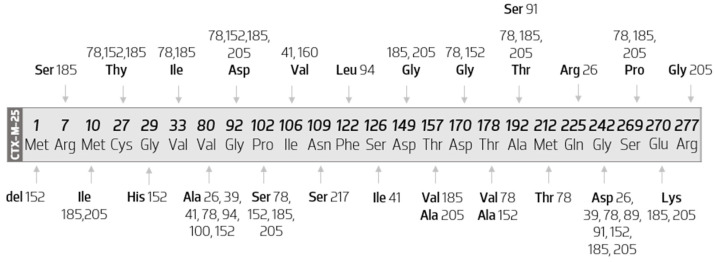
Diagram map of the substitutions that occur within CTX-M-25 cluster. Scheme based on available sequences and phenotypes in the Bacterial Antimicrobial Resistance Reference Gene Database.

**Table 1 biology-11-00256-t001:** Distribution of CTX-M-1-like enzymes by major hotspot mutation sites. M1 to M5 represent the five hotspots in CTX-M-1 cluster; For each hotspot, the substitution of the respective amino acid is represented; The numbers for each hotspot represent the indexes of all the enzymes from the CTX-M-1 cluster that present the corresponding hotspot.

Hotspot	CTX-M-1 Variants
Val80Ala (MI)	3, 10, 12, 15, 22, 28, 29, 30, 33, 34, 36, 37, 42, 54, 62, 64, 66, 68, 71, 72, 80, 82, 88, 96, 101, 103, 117, 123, 127, 132, 139, 143, 150, 154, 155, 156, 157, 162, 163, 167, 169, 170, 172, 173, 176, 177, 178, 180, 181, 182, 183, 184, 186, 187, 188, 189, 193, 194, 197, 199, 202, 203, 204, 206, 208, 209, 210, 211, 216, 218, 220, 224, 225, 228, 231, 232
Asp117Asn (M2)	3, 10, 12, 15, 22, 23, 28, 32, 34, 42, 52, 53, 54, 57, 60, 62, 64, 66, 68, 69, 71, 72, 79, 80, 82, 88, 96, 101, 103, 114, 116, 117, 123, 127, 132, 136, 138, 139, 142, 144, 150, 154, 155, 156, 157, 162, 163, 164, 167, 169, 170, 172, 173, 176, 177, 178, 179, 180, 181, 182, 183, 184, 186, 187, 188, 189, 190, 193, 194, 197, 199, 202, 203, 204, 206, 207, 208, 209, 210, 211, 212, 216, 218, 220, 224, 225, 226, 228, 230, 231, 232
Ser143Ala (M3)	3, 10, 12, 15, 22, 23, 28, 29, 30, 33, 34, 37, 42, 52, 53, 54, 57, 60, 62, 64, 66, 68, 69, 71, 72, 79, 80, 82, 88, 96, 101, 103, 114, 116, 117, 123, 132, 136, 139, 142, 143, 144, 150, 155, 156, 157, 162, 163, 164, 167, 169, 170, 172, 173, 176, 177, 178, 179, 180, 181, 182, 183, 184, 186, 187, 188, 189, 190, 193, 194, 197, 199, 202, 203, 204, 206, 207, 208, 209, 210, 211, 212, 216, 218, 220, 224, 225, 226, 227, 228, 230, 231, 232
Asp242Gly (M4)	15, 28, 29, 32, 33, 53, 64, 69, 71, 79, 82, 88, 96, 101, 103, 114, 117, 123, 127, 132, 139, 142, 143, 144, 150, 154, 155, 156, 157, 163, 164, 169, 170, 172, 173, 176, 178, 179, 180, 181, 182, 183, 184, 186, 188, 189, 190, 193, 194, 197, 199, 202, 208, 209, 210, 216, 218, 224, 225, 226, 227, 228, 230, 231, 232
Asn289Asp (M5)	3, 10, 12, 15, 29, 30, 33, 34, 37, 42, 52, 53, 54, 57, 60, 61, 62, 64, 66, 68, 71, 72, 80, 82, 88, 96, 101, 103, 114, 117, 123, 132, 136, 139, 142, 143, 144, 150, 155, 156, 157, 162, 163, 164, 167, 169, 170, 172, 173, 176, 177, 178, 179, 180, 181, 182, 183, 184, 186, 187, 188, 189, 190, 193, 194, 197, 199, 202, 203, 204, 206, 207, 208, 209, 210, 211, 212, 216, 218, 220, 224, 225, 226, 227, 228, 230, 231, 232

**Table 2 biology-11-00256-t002:** List of variant-specific mutations among CTX-M-1 cluster enzymes.

Enzyme	Mutation	Enzyme	Mutation	Enzyme	Mutation
CTX-M-54	Pro170Gln	CTX-M-143	Asn173Ser	CTX-M-194	Arg7His
CTX-M-64	Ser89Thr	CTX-M-154	Glu39Lys	CTX-M-197	Leu158Val
CTX-M-66	Ser23Asn	CTX-M-164	Gln8Lys	CTX-M-204	Thr168Ile
CTX-M-68	Glu161Asp	CTX-M-170	Asp203Asn	CTX-M-207	Asp242Ser
CTX-M-72	Arg167Gly	CTX-M-172	Leu141Pro	CTX-M-208	Ala222Glu
CTX-M-88	Arg277His	CTX-M-173	Ala153Thr	CTX-M-209	Thr212Ile
CTX-M-101	Ser126Ile	CTX-M-176	Thr267Ala	CTX-M-210	Ala31Thr
CTX-M-103	Ser240Asn	CTX-M-177	Gln157Leu	CTX-M-211	Met2Lys
CTX-M-114	Val77Ala	CTX-M-179	Pro269Thr		Val2Ser
CTX-M-117	Pro177Gln	CTX-M-181	Asn135His	CTX-M-216	Gly290Ser
CTX-M-123	Val233Thr	CTX-M-182	Lys36Arg	CTX-M-224	Gln8Leu
CTX-M-127	Asn138Thr	CTX-M-183	Thr30Lys	CTX-M-226	Pro25Thr
CTX-M-138	Ile286Ser	CTX-M-184	Asn58Thr	CTX-M-227	Gly290Arg
CTX-M-139	Tyr27Phe	CTX-M-186	Thr20Ser	CTX-M-228	Ser223Asn
CTX-M-142	Asp32Asn	CTX-M-187	Leu291Ser	CTX-M-230	Ala174Ile
CTX-M-150	Asn216His	CTX-M-188	Ala82Thr	CTX-M-231	Ala31Glu
	Thr218His	CTX-M-189	Ser133Gly	CTX-M-232	Ala15Thr

**Table 3 biology-11-00256-t003:** Enzymes from CTX-M-9 cluster enzymes and their respective exclusive mutations.

Enzyme	Mutation	Enzyme	Mutation
CTX-M-17	Glu289Lys	CTX-M-104	Ser275Asn
CTX-M-21	Ala13Gly	CTX-M-110	Lys114Glu
	Ala14Gly		Leu291Asn
	Cys16Gly	CTX-M-111	Pro148Gln
	Leu26Phe	CTX-M-112	Ser126Gly
	Val33Gly	CTX-M-113	Gln86Arg
CTX-M-38	Ser223Arg	CTX-M-126	Pro170Ala
CTX-M-73	Val33Met	CTX-M-134	Asp242Ser
CTX-M-81	Lys85Glu	CTX-M-161	Gly149Glu
	Cys101Gln	CTX-M-168	Pro170His
CTX-M-83	Gln60His	CTX-M-198	Pro25Ala
CTX-M-84	Thr211Ala	CTX-M-201	Asn107Ser
CTX-M-85	Leu122Pro	CTX-M-213	Ala156Glu
CTX-M-86	Ile111Phe		

## Data Availability

Data sharing is not applicable to this article.

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
