# Peer review of "New CTX-M Group Conferring β-Lactam Resistance: A Compendium of Phylogenetic Insights from Biochemical, Molecular, and Structural Biology"

_biology, 2022, doi:10.3390/biology11020256_

Round 1

Reviewer 1 Report

Jacinta et al., did clustering for CTX-M β-lactamases that are very important in clinics. Even though it may need a certain amount of analysis, but it is slightly wondering of this kind of work that was done with compuational analysis without testing actual bacteria. If the authors could've added some real biochemical data, together with computational analysis, it would be better. And there is no suggestion about a strategy what things to do based on this clustering. It may be more useful for readers to get discussed on what to do with this data.  

Author Response

Authors: It is our pleasure to submit the revised version of the manuscript for consideration in Biology journal. In this new version we took upon the reviewers’ comments and suggestions and changed the content and organization of the manuscript. Overall we believe this new version meets our goals for this work as well as the reviewers’ suggestions.  

#1

Jacinta et al., did clustering for CTX-M β-lactamases that are very important in clinics. Even though it may need a certain amount of analysis, but it is slightly wondering of this kind of work that was done with computational analysis without testing actual bacteria. If the authors could've added some real biochemical data, together with computational analysis, it would be better. And there is no suggestion about a strategy what things to do based on this clustering. It may be more useful for readers to get discussed on what to do with this data. 

Authors: We kindly thank you for the insights and suggestions. Like mentioned in the new version of the manuscript, due to previous speculations regarding CTX-M-151 as a new CTX-M’s group, we took upon the task of performing an update on phylogeny and molecular structure analysis. Looking into the literature, a recent study conducted by Ghiglione et al provided experimental insights regarding this β-lactamase, particularly its pharmacological profile. According to the authors, CTX-M-151 presents a distinct resistance pattern from the other CTX-Ms. These results corroborate our data and proposal to classify CTX-M-151 as a distinct sixth group among the β-lactamase enzymes. Moreover, we believe this update shows the relevance of ongoing research in this field, considering CTX-Ms increasing fast pharmacological adaptation.

Reviewer 2 Report

The manuscript describes a bioinformatic analysis of CTX-M sequences deposited in the NCBI database.  The text is difficult to follow in many places and needs stringent editing of spelling and basic English grammar to improve its clarity. The description of the phylogenetic analysis lacks detail and further details about the method, any assumptions built into the analysis and statistical quality checks need to be included.  The principal conclusion, that CTX-M-151 may represent a significantly different lineage confirms previous publication without adding new detail.

There are numerous points to be addressed,  including:

P.1 l. 33 Spelöling of Alexander Fleming.  Penicllin was not the first antibiotic (sulphonamides were), and he only described its effect - the discovery of the molecular entity came later.

P.2. l. 47 and possibly elsewhere, Gram-negative should always have capital G.

P. 2. l. 76 K. georgiana

P.3. l. 97. "enzymes belonging to in this database"

P. 4. l. 131. "multiple sequence analysis"

P. 6. l. 160 "substituions" not "substations"

P.6. Fig. 3 The structures are too small to be useful.  Could superimposed ribbon diagrams be shown instead, perhaps highlighting the areas of significant difference between the structures?

Author Response

Authors (A): It is our pleasure to submit the revised version of the manuscript for consideration in Biology journal. In this new version we took upon the reviewers’ comments and suggestions and changed the content and organization of the manuscript. Overall we believe this new version meets our goals for this work as well as the reviewers’ suggestions.  

#2

The manuscript describes a bioinformatic analysis of CTX-M sequences deposited in the NCBI database.  The text is difficult to follow in many places and needs stringent editing of spelling and basic English grammar to improve its clarity. The description of the phylogenetic analysis lacks detail and further details about the method, any assumptions built into the analysis and statistical quality checks need to be included.  The principal conclusion, that CTX-M-151 may represent a significantly different lineage confirms previous publication without adding new detail.

There are numerous points to be addressed, including:

P.1 l. 33 Spelöling of Alexander Fleming. Penicllin was not the first antibiotic (sulphonamides were), and he only described its effect - the discovery of the molecular entity came later.

A.1. We thank you for your corrections and the appropriate corrections are now updated

P.2. l. 47 and possibly elsewhere, Gram-negative should always have capital G.

  1. 3. l. 76 K. georgiana

P.4. l. 97. "enzymes belonging to in this database"

  1. l. 131. "multiple sequence analysis"
  2. l. 160 "substituions" not "substations"

A.6. We kindly thank you for these corrections. They represented oversights that are now properly revised

P.7. Fig. 3 The structures are too small to be useful.  Could superimposed ribbon diagrams be shown instead, perhaps highlighting the areas of significant difference between the structures?

A.7. We thank you for your insight. Taking the size and complexity of the different structures in question, to overlap them would not allow to assess the structural differences among them, in our opinion. The purpose of Figure 3 was to show that, despite the significant sequence differences among the different groups showed in Figure 2, visually all of them appear somewhat similar. However, in order to meet the suggestions, we have simplified Figure 3. It now highlights the two different enzyme lengths among all CTX-Ms, with CTX-M-1 as the representative of CTX-Ms with 291 amino acids and CTX-M-151 as the enzyme with 304 amino acids. We believe that this edited image allows to show, simultaneously, that CTX-M-151 distinguishes itself from the other CTX-Ms, while showing that, despite the differences, there are structural similarities.

Round 2

Reviewer 1 Report

It seems to be sufficient in publishing in this journal.

Author Response

We would like to, once again, thank the reviewer for the comments and insights. In this updated version we believe the suggestions given were once again taken into consideration and hope this version meets the goals on both parties.  

Revisor #2

The revised manuscript is still difficult to read and, although some discussion about the methods has been added, the description remains brief.  The manuscript would benefit from further editing to improve its clarity.  There are numerous points to be addressed, including:

We kindly acknowledge the several errors and they have now been properly corrected accordingly.

  1. 38: Penicillin is mentioned twice, one occurrence is redundant. The sentence is misleading. It ignores the sulfa drugs that were in use (and saving lives) before penicillin was introduced in the 1940's. References 1-3 are not appropriate for the historical subject of this sentence and should be replaced.

A – Q1: The sentence as now been re-written and references have been updated. The purpose of the sentence is to highlight the profound impact that penicillin had in the history of infections. Hence the stated “…a turning point…” rather than “…the turning point…”. We believe that the sentence doesn’t disregard the importance of previous relevant drugs.

  1. 40. "portraying" is not the right word. Perhaps the authors mean "foreshadowing", as the discovery of penicillinase predated the clinical use of penicillin.

A – Q2: Done.

  1. 44: "alerting" does not seem right in this context.

A– Q3: Done.

  1. 48: the plural "beta-lactams" cannot be "one".

A – Q4: Done.

  1. 54: Why "concomitantly"? Changed to “as a result”

A – Q5: Done.

  1. 119: "CTX-M beta-lactamases"

A – Q6: Done.

  1. 121: "as well as and " redundancy.

A – Q7: Done.

  1. 137: "multiple sequence analysis", not "sequencing".

A – Q8: Done.

  1. 150 and similar in many places: "amino acids substitution" should be "amino acid substitutions".

A – Q9: Done.

  1. 151 and elsewhere: "three dimensional" not "tridimensional".

A – Q10: Done.

  1. 184 and elsewhere: why is "wild type" in italics (and in this case, green)? Delete.

A – Q11: Done.

  1. 196 and elsewhere similar in other clusters: "within the CTX-M-2 cluster" rather than "within CTX-M-2 cluster’s members".

A – Q12. Done.

  1. 68: What "functional heterogeneity"? Nothing has been said about function of the enzymes, only a discussion of sequences. In the preceding paragraph, we are told that the enzymes all belong to the same group of the Bush et al. functional classification, which implies homogeneity of function. In general, more attention should be given to the relevance of the sequence differences to enzymatic properties (and, therefore, resistance).

A – Q13. We acknowledge the comment. We agree with the reviewer and the sentence has now been re-written. “This molecular heterogeneity assumes clinical relevance, since it may be responsible for its resistance profile.”

  1. 84: Although it is indicated as a focus, nothing is said about "enzymatic properties" of the various clusters that are identified by the sequence analysis.

A – Q14. The question regarding “enzymatic properties” is quite vague. We should not mix functional classification with structural classification schemes. Nevertheless, if we consider the affinity and other kinetic properties (Km, Kcat Vmax, Km/Vmax, …) of the now proposed 6 clusters of CTX-Ms towards the different antimicrobials (with cefotaxime, ceftazidime, ceftriaxone being a small example of the dozens of molecules to be mentioned of) as described by Ghiglione and collaborators (2021) just for the novel CTX-M-151 it would be endless and so unpracticable, in our perspective. The main reason is that there is no kinetic constancy since each “enzyme properties” varies enormously between each individual enzyme (independently of its own group). 

  1. 158 onwards and Fig. 3. The interpretation of the 3D structures remains difficult and unconvincing. Although the two molecules in Fig. 3 have not been superimposed to present them in identical orientation (here overlapping ribbons would be more appropriate), it is clear that there are significant differences and "incredible similarities" is inappropriate.

A – Q15. We acknowledge the comments regarding the pertinence of Figure 3. As previously mentioned, the purpose of Figure 3 was to show that, despite the significant sequence differences, 163 amino acids in a total of 304 for CTX-M-151 and the extra 13 residues, the 3D structures visually appear somewhat similar. However, we understand the reviewer’s vision and, to meet it, we have overlapped CTX-M-1 and CTX-M-151 enzymes, in order to demonstrate the mentioned structural similarities. We hope that the edited Figure allows the authors’ goal to come across successfully.

  1. 171. "CTX-M-1 wild type gene" delete "wild type". It is inappropriate, CTX-M-1 was the first sequence described, not necessarily the progenitor of all subsequent variants. The same applies to further references to "wild type" in the subsequent discussion of other clusters.

A – Q16. We acknowledge the comment and we have now removed the term “wild type” in order to avoid misleading statements.

Line. 259: Section 4.2 is not a very useful discussion, most of it is devoted to cataloguing of amino acid substitutions that (is necessary) belongs in the results section.  A more detailed discussion of the structural and mechanistic differences between CTX-M 151 and other CTX-M enzymes is needed if this section is to be retained.  Which are the residues in the different clusters that are "structural requirements of β-lactamases CTX-M, necessary for the presence of their typical profiles"? It is a pity that there is no discussion of the respective enzymatic properties of representative enzymes in the various clusters.

A – Q17. Although we believe that the cataloguing section is important and useful, we understand how its detailed description may be difficult to read. As such, and taking upon the reviewer’s point of view, we have moved the detailed intra-cluster analysis to the results section.  Regarding the mechanistic (functional?) properties/differences, and accordingly with the answer to question 14 (A – Q14), describing such enzyme properties would represent an endless amount of functional (kinetic/mechanistic) information since all enzymes behaved independently of their structural cluster.

Reviewer 2 Report

The revised manuscript is still difficult to read and, although some discussion about the methods has been added, the description remains brief.  The manuscript would benefit from further editing to improve its clarity.  

There are too many errors to list in detail, but a few points to be addressed include:

L. 38: Penicillin is mentioned twice, one occurrence is redundant. The sentence is misleading. It ignores the sulfa drugs that were in use (and saving lives) before penicillin was introduced in the 1940's. References 1-3 are not appropriate for the historical subject of this sentence and should be replaced.

L. 40. "portraying" is not the right word. Perhaps the authors mean "foreshadowing", as the discovery of penicillinase predated the clinical use of penicillin.

L. 44: "alerting" does not seem right in this context.

L. 48: the plural "beta-lactams" cannot be "one".

L. 54: Why "concomitantly"?

L. 68: What "functional heterogeneity"? Nothing has been said about function of the enzymes, only a discussion of sequences.  In the preceding paragraph, we are told that the enzymes all belong to the same group of the Bush et al. functional classification, which implies homogeneity of function.  In general, more attention should be given to the relevance of the sequence differences to enzymatic properties (and, therefore, resistance).

L. 84: Although it is indicated as a focus, nothing is said about "enzymatic properties" of the various clusters that are identified by the sequence analysis.  

L. 119: "CTX-M beta-lactamases"

L. 121: "as well as and " redundancy.

L. 137: "multiple sequence analysis", not "sequencing".

L. 150 and similar in many places: "amino acids substitution" should be "amino acid substitutions".

 L. 151 and elsewhere: "three dimensional" not "tridimensional".

L. 158 onwards and Fig. 3.  The interpretation of the 3D structures remains difficult and unconvincing. Although the two molecules in Fig. 3 have not been superimposed to present them in identical orientation (here overlapping ribbons would be more appropriate), it is clear that there are significant differences and "incredible similarities" is inappropriate.

L. 171. "CTX-M-1 wild type gene" delete  "wild type". It is inappropriate, CTX-M-1 was the first sequence described, not necessarily the progenitor of all subsequent variants. The same applies to further references to "wild type" in the subsequent discussion of other clusters.

L. 184 and elsewhere: why is "wild type" in italics (and in this case, green)? Delete.

L. 196 and elsewhere similar in other clusters: "within the CTX-M-2 cluster" rather than "within CTX-M-2 cluster’s members".   

Line. 259: Section 4.2 is not a very useful discussion, most of it is devoted to cataloguing of amino acid substitutions that (is necessary) belongs in the results section.  A more detailed discussion of the structural and mechanistic differences between CTX-M 151 and other CTX-M enzymes is needed if this section is to be retained.  Which are the residues in the different clusters that are "structural requirements of β-lactamases CTX-M, necessary for the presence of their typical profiles"? It is a pity that there is no discussion of the respective enzymatic properties of representative enzymes in the various clusters. 

Author Response

We would like to, once again, thank the reviewer for the comments and insights. In this updated version we believe the suggestions given were once again taken into consideration and hope this version meets the goals on both parties.

Revisor #2

The revised manuscript is still difficult to read and, although some discussion about the methods has been added, the description remains brief. The manuscript would benefit from further editing to improve its clarity. There are numerous points to be addressed, including:

A: We kindly acknowledge the several errors and they have now been properly corrected accordingly.

  1. 38: Penicillin is mentioned twice, one occurrence is redundant. The sentence is misleading. It ignores the sulfa drugs that were in use (and saving lives) before penicillin was introduced in the 1940's. References 1-3 are not appropriate for the historical subject of this sentence and should be replaced.

A – Q1: The sentence as now been re-written and references have been updated. The purpose of the sentence is to highlight the profound impact that penicillin had in the history of infections. Hence the stated “…a turning point…” rather than “…the turning point…”. We believe that the sentence doesn’t disregard the importance of previous relevant drugs.

  1. 40. "portraying" is not the right word. Perhaps the authors mean "foreshadowing", as the discovery of penicillinase predated the clinical use of penicillin.

A – Q2: Done.

  1. 44: "alerting" does not seem right in this context.

A– Q3: Done.

  1. 48: the plural "beta-lactams" cannot be "one".

A – Q4: Done.

  1. 54: Why "concomitantly"? Changed to “as a result”

A – Q5: Done.

  1. 119: "CTX-M beta-lactamases"

A – Q6: Done.

  1. 121: "as well as and " redundancy.

A – Q7: Done.

  1. 137: "multiple sequence analysis", not "sequencing".

A – Q8: Done.

  1. 150 and similar in many places: "amino acids substitution" should be "amino acid substitutions".

A – Q9: Done.

  1. 151 and elsewhere: "three dimensional" not "tridimensional".

A – Q10: Done.

  1. 184 and elsewhere: why is "wild type" in italics (and in this case, green)? Delete.

A – Q11: Done.

  1. 196 and elsewhere similar in other clusters: "within the CTX-M-2 cluster" rather than "within CTX-M-2 cluster’s members".

A – Q12. Done.

  1. 68: What "functional heterogeneity"? Nothing has been said about function of the enzymes, only a discussion of sequences. In the preceding paragraph, we are told that the enzymes all belong to the same group of the Bush et al. functional classification, which implies homogeneity of function. In general, more attention should be given to the relevance of the sequence differences to enzymatic properties (and, therefore, resistance).

A – Q13. We acknowledge the comment. We agree with the reviewer and the sentence has now been re-written. “This molecular heterogeneity assumes clinical relevance, since it may be responsible for its resistance profile.”

  1. 84: Although it is indicated as a focus, nothing is said about "enzymatic properties" of the various clusters that are identified by the sequence analysis.

A – Q14. The question regarding “enzymatic properties” is quite vague. We should not mix functional classification with structural classification schemes. Nevertheless, if we consider the affinity and other kinetic properties (Km, Kcat Vmax, Km/Vmax, …) of the now proposed 6 clusters of CTX-Ms towards the different antimicrobials (with cefotaxime, ceftazidime, ceftriaxone being a small example of the dozens of molecules to be mentioned of) as described by Ghiglione and collaborators (2021) just for the novel CTX-M-151 it would be endless and so unpracticable, in our perspective. The main reason is that there is no kinetic constancy since each “enzyme properties” varies enormously between each individual enzyme (independently of its own group).

  1. 158 onwards and Fig. 3. The interpretation of the 3D structures remains difficult and unconvincing. Although the two molecules in Fig. 3 have not been superimposed to present them in identical orientation (here overlapping ribbons would be more appropriate), it is clear that there are significant differences and "incredible similarities" is inappropriate.

A – Q15. We acknowledge the comments regarding the pertinence of Figure 3. As previously mentioned, the purpose of Figure 3 was to show that, despite the significant sequence differences, 163 amino acids in a total of 304 for CTX-M-151 and the extra 13 residues, the 3D structures visually appear somewhat similar. However, we understand the reviewer’s vision and, to meet it, we have overlapped CTX-M-1 and CTX-M-151 enzymes, in order to demonstrate the mentioned structural similarities. We hope that the edited Figure allows the authors’ goal to come across successfully.

  1. 171. "CTX-M-1 wild type gene" delete "wild type". It is inappropriate, CTX-M-1 was the first sequence described, not necessarily the progenitor of all subsequent variants. The same applies to further references to "wild type" in the subsequent discussion of other clusters.

A – Q16. We acknowledge the comment and we have now removed the term “wild type” in order to avoid misleading statements.

Line. 259: Section 4.2 is not a very useful discussion, most of it is devoted to cataloguing of amino acid substitutions that (is necessary) belongs in the results section. A more detailed discussion of the structural and mechanistic differences between CTX-M 151 and other CTX-M enzymes is needed if this section is to be retained. Which are the residues in the different clusters that are "structural requirements of β-lactamases CTX-M, necessary for the presence of their typical profiles"? It is a pity that there is no discussion of the respective enzymatic properties of representative enzymes in the various clusters.

A – Q17. Although we believe that the cataloguing section is important and useful, we understand how its detailed description may be difficult to read. As such, and taking upon the reviewer’s point of view, we have moved the detailed intra-cluster analysis to the results section. Regarding the mechanistic (functional?) properties/differences, and accordingly with the answer to question 14 (A – Q14), describing such enzyme properties would represent an endless amount of functional (kinetic/mechanistic) information since all enzymes behaved independently of their structural cluster.

Round 3

Reviewer 2 Report

The manuscript has been significantly improved and can be accepted.

Author Response

We would like to thank the reviewer for the comments.